# Quantifying soil accumulation of atmospheric mercury using fallout radionuclide chronometry

Joshua D. Landis [1] ✉, Daniel Obrist[2,3], Jun Zhou [4], Carl E. Renshaw[1], William H. McDowell [5,6], Christopher J. Nytch[7], Marisa C. Palucis[1], Joanmarie Del Vecchio[8], Fernando Montano Lopez [9] & Vivien F. Taylor[1]

Soils are a principal global reservoir of mercury (Hg), a neurotoxic pollutant that is accumulating through anthropogenic emissions to the atmosphere and subsequent deposition to terrestrial ecosystems. The fate of Hg in global soils remains uncertain, however, particularly to what degree Hg is re-emitted back to the atmosphere as gaseous elemental mercury (GEM). Here we use fallout radionuclide (FRN) chronometry to directly measure Hg accumulation rates in soils. By comparing these rates with measured atmospheric fluxes in a mass balance approach, we show that representative Arctic, boreal, temperate, and tropical soils are quantitatively efficient at retaining anthropogenic Hg. Potential for significant GEM re-emission appears limited to a minority of coniferous soils, calling into question global models that assume strong re-emission of legacy Hg from soils. FRN chronometry poses a powerful tool to reconstruct terrestrial Hg accumulation across larger spatial scales than previously possible, while offering insights into the susceptibility of Hg mobilization from different soil environments.

Mercury (Hg) is a neurotoxic heavy metal, emitted to the atmosphere by both anthropogenic activities and natural sources and dispersed globally through atmospheric transport[1]. Forest ecosystems are the largest global sink of atmospheric Hg due to foliar uptake of gaseous elemental mercury (GEM) from the atmosphere[2]. Hg is subsequently cycled to underlying soils by litterfall (LF), canopy throughfall (TF), and additional deposition of GEM directly to the forest floor and other above-ground surfaces[2–6]. After centuries of anthropogenic Hg emissions, soils represent the largest global reservoir of anthropogenic Hg and pose an ongoing risk of exposure to terrestrial and aquatic food webs through uptake, leaching, and erosive transport to aquatic ecosystems[2,4,7–10]. In a biogeochemical twist, soil Hg is also partially re-emitted back to the

atmosphere. Such re-emission of GEM counteracts the migration of Hg from surface to deep mineral soils through pedogenic processes and thus prevents Hg from entering long-term sequestration[11–16]. The scale of terrestrial GEM re-emission remains highly uncertain, however, with global estimates spanning 1000–3000 Mg y⁻¹, or roughly 30–100% of gross terrestrial Hg deposition[4,7,12,17]. This uncertainty raises fundamental questions about the long-term efficacy of soils in sequestering legacy and contemporary Hg, the lifetimes of Hg in the surface environment, and as a result about the fate of Hg in the context of a changing global environment[1,3,4,6].

Contemporary measurements of GEM fluxes at the soil surface reveal complex bi-directional exchanges between soils and the

[1]Department of Earth Sciences, Dartmouth College, Hanover, NH 03755, USA. [2]Department of Environmental, Earth, and Atmospheric Sciences, University of Massachusetts, Lowell, MA 01854, USA. [3]Division of Agriculture and Natural Resources, University of California, Davis, CA 95616, USA. [4]State Key Laboratory of Soil and Sustainable Agriculture, Institute of Soil Science, Chinese Academy of Sciences, Nanjing 210008, China. [5]Department of Natural Resources and the Environment, University of New Hampshire, Durham, NH, USA. [6]Institute of Environment, Florida International University, Miami, FL, USA. [7]Department of Environmental Sciences, University of Puerto Rico – Rio Piedras, San Juan, PR 00925, USA. [8]Department of Geology, William and Mary, Williamsburg, VA 23188, USA. [9]Department of Biological Sciences, Dartmouth College, Hanover, NH 03755, USA. ✉e-mail: joshua.d.landis@dartmouth.edu

atmosphere, with evidence for both net deposition and net emission (Fig. 1)[3,13,14,18,19]. Further complicating Hg dynamics, ecosystem components can play opposing roles in Hg cycling; for example, foliage acts as a strong and unambiguous net sink[4,20], but forest floors (leaf litter or soil Oi horizon) may act as either source or sink[19,21,22]. Within the soil profile, high pore-gas GEM concentrations in some surface soils point to leaf litter and soil humus as net emitters of GEM during the decomposition of organic matter, which we term legacy emission if derived from past pollution. In deeper soils, GEM concentrations are lower than atmospheric levels indicating mineral soils are a stable Hg sink[14,21,23–25]. The balance of these two pathways has profound influences on the global Hg cycle. Processes that favor Hg retention in soils effectively remove Hg pollution from continued circulation in the environment, while legacy emissions extend exposures to pollution and globally delay recovery from Hg pollution even as primary anthropogenic Hg emissions continue to decrease due to national and international legislative actions[26].

Recent advancements in soil chronometry with atmospheric fallout radionuclides (FRNs) provide an exciting opportunity to directly measure Hg soil accumulation rates using a "bottom–up" mass balance approach, thereby providing a critical missing view of terrestrial Hg biogeochemistry. Already the FRN ²¹⁰Pb (half-life 22.3 years) is a vital tool for reconstructing historical Hg deposition using both lake sediment and peat environmental archives[27–29]. To extend this approach to terrestrial ecosystems, we employed and validated FRN age models that accurately date foliar and soil organic matter (SOM). Three independent radionuclide age models, the Linked Radionuclide aCcumulation (LRC) model based on ⁷Be:²¹⁰Pb, the nuclear bomb-pulse ²⁴¹Am, and the novel ²²⁸Th:²²⁸Ra chronometer, yield concordant SOM ages spanning subannual to centennial timescales[30–32]. The concordance of the models is predicated on the strong affinity of these five particle-reactive metals for SOM through the formation of stable organometallic complexes. The migration rate of ²¹⁰Pb in soils is identical to that of atmospheric ¹⁴C[33], confirming that sorption of the tracer metals to SOM persists from uptake in vegetation through incorporation in soil, subsequent SOM decomposition, and hydrologic transport in association with natural organic matter[33–38]. The FRN chronometers accurately measure soil organometallic dynamics not possible with more conventional Δ¹⁴C chronometry due to uncertain lag times between $CO_2$ assimilation and SOM production in the forest carbon cycle[39–43].

With their ability to accurately date SOM, the chronometer metals provide a powerful hypothesis for predicting Hg dynamics in soils, and thus an emerging tool for assessing the extent to which Hg behaves as a conservative particle-reactive metal similarly entrained in the pedogenic cycle with SOM versus a fraction that may be re-emitted or otherwise mobilized as GEM. Here we employ LRC and ²⁴¹Am age models to measure Hg accumulation rates in a range of soils spanning Arctic, boreal, temperate, and tropical ecosystems. A subset of soils for which Hg is reported here were also previously dated by the ²²⁸Th:²²⁸Ra method[32]. We then directly compare rates of soil Hg accumulation over annual to centennial timescales with measured whole-ecosystem atmospheric depositional fluxes. This approach thus provides a new basis for constraining the fate of past soil Hg accumulation, for understanding key processes that regulate Hg accumulation in soils, and for assessing the susceptibility of soil Hg to legacy emission and redistribution in response to environmental and climate perturbation.

## Results and discussion
### Soil ages and Hg accumulation rates
FRN age estimates provide a critical and objective basis for comparing Hg accumulation in foliage and organic soil O-horizons across contrasting ecosystems (Fig. 2, Table S1). We define soil Oi horizon as senesced leaves on the forest floor; Oe horizon as partially decomposed leaf fragments and fibers; and Oa horizon as humified organic matter lacking identifiable leaf fragments. Within each ecosystem, ages increase from foliage, Oi, Oe, and Oa to reflect clear differentiation of horizons as they were sampled in the field [$p < 0.05$]. Ages pooled across ecosystems increase significantly from 1.5 years in foliage to 3.3 years in Oi horizons, 7.5 years in Oe horizons, and 29 years in Oa horizons. Ages of Arctic foliage and soil horizons are significantly older than corresponding temperate, boreal, or tropical horizons which we attribute to slower SOM decomposition at high latitudes [$p < 0.05$]. Similarly, tropical soil horizons are significantly younger than corresponding temperate ones [$p < 0.05$]. There were no significant differences between temperate ($n = 23$ sites) and boreal forests ($n = 4$ sites), and these are hereafter analyzed together. Note that data in Fig. 2 represent sampled depth intervals and not entire horizons.

Total Hg concentrations pooled across ecosystems increase significantly from 41 ng g⁻¹ in foliage to 74 ng g⁻¹ in Oi, 140 ng g⁻¹ in Oe, and 192 ng g⁻¹ in Oa horizons (Fig. 2b; Table S1). There were no significant differences in foliar concentrations among ecosystems [$p < 0.05$]. Only Arctic Oi and temperate Oa Hg concentrations are significantly higher than contrasting ecosystems [$p < 0.05$], but concentration data are poor predictors of Hg accumulation rates. Soil Hg concentrations were converted to total inventories by multiplication with total soil mass recovered from the defined area of quantitative soil pits (per cm thickness, g m⁻² cm⁻¹) (Fig. 2c). Oa horizons store the largest soil Hg inventories in each ecosystem, increasing significantly from 140 µg m⁻² cm⁻¹ in tropical soils, 440 µg m⁻² cm⁻¹ in temperate deciduous soils, 470 µg m⁻² cm⁻¹ in temperate coniferous soils, and

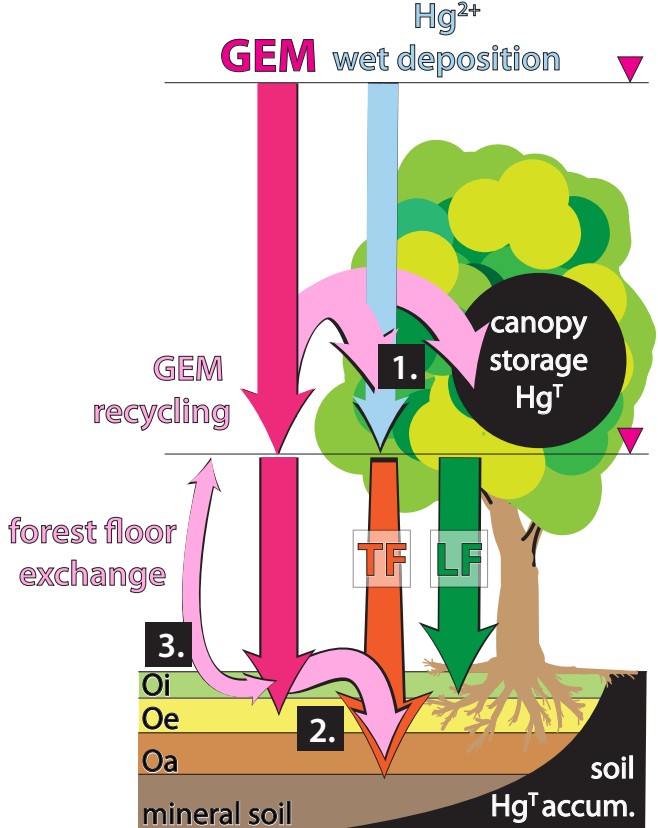

**Fig. 1 | Conceptual model of atmospheric gaseous elemental mercury (GEM) dynamics in soil Hg accumulation.** GEM is assimilated by foliage and enters soils through litterfall (LF). Additional pathways include the rinsing of GEM from the canopy by throughfall (TF) [1] and mobilization of GEM from the forest floor to underlying mineral soil [2]. GEM may also be re-emitted from the forest floor back to the atmosphere [3] from where it may be either recycled back through TF and LF or exported to the global atmospheric pool.

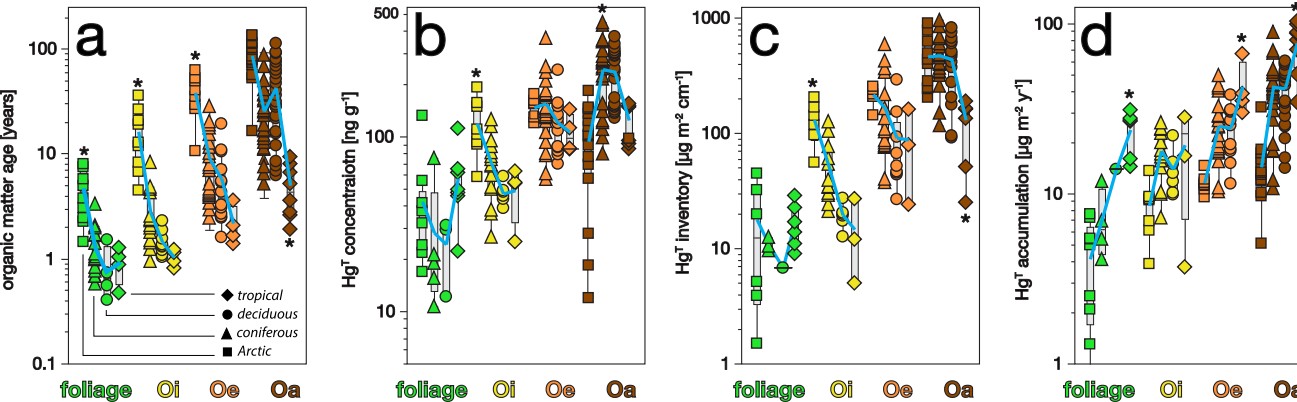

**Fig. 2 | Soil ages and Hg compared by horizon across Arctic, temperate/boreal deciduous and coniferous, and tropical ecosystems. a** Soil organic matter age (LRC age model); green = foliage; yellow = Oi horizon; orange = Oe, brown = Oa. **b** Soil total Hg concentrations; these are not corrected for geogenic contributions, see text. **c** Total Hg inventories cm$^{-1}$ horizon depth. **d** Hg accumulation rates versus horizon cm$^{-1}$ depth interval. Box plots indicate medians and 25%, 50%, and 75% quartiles. Asterisk (*) marks ecosystem comparisons that are significantly different across horizons [ANOVA, $p < 0.05$]. Blue lines connect horizon mean values across ecosystems.

460 µg m$^{-2}$ cm$^{-1}$ in Arctic soils [$p < 0.05$]. Previously, high Hg concentrations and inventories in Oa horizons have been attributed to their older ages and longer accumulation times of this horizon, but these studies lacked the ability to comprehensively date soils layer by layer[44]. For reference, total atmospheric Hg stocks in full soil profiles increased significantly across Arctic, temperate, and tropical ecosystems, from $2.4 \pm 0.4$ mg m$^{-2}$ (mean ± SD, $n = 4$), to $6.9 \pm 3.4$ mg m$^{-2}$ ($n = 10$), and $11.2 \pm 2.4$ mg m$^{-2}$ ($n = 3$), respectively [$p < 0.05$; Table S2]. Hg inventories were summed through mineral soils to depths of up to 50 cm corresponding in each profile to calendar years >1900 and were corrected for geogenic contribution (see "Methods").

Rates of foliar and soil Hg accumulation are converted from Hg inventories using LRC soil ages. Accumulation rates increase substantially and statistically significantly across ecosystems: Arctic < temperate < tropical [$p < 0.05$; Fig. 2d]. Temperate foliar Hg accumulation estimates of $12 \pm 3$ µg m$^{-2}$ y$^{-1}$ (mean ± SE, $n = 4$) are in good agreement with adjacent LF data for three forest sites (average of Harvard, Howland, and Downer Forests = $11.8 \pm 0.8$ µg m$^{-2}$ y$^{-1}$), confirming that the LRC approach accurately reconstructs foliar Hg deposition rates. Arctic foliar fluxes average $7.5 \pm 2.9$ µg m$^{-2}$ y$^{-1}$ ($n = 4$), comparable with prior estimates and consistent with lower foliar-derived Hg deposition in the high latitudes[45–47]. Tropical foliar fluxes average $28 \pm 3$ µg m$^{-2}$ y$^{-1}$ when considering only the dominant canopy but increase to $49 \pm 9$ µg m$^{-2}$ y$^{-1}$ when significant understory vegetation is also included ($n = 3$). Foliar Hg deposition fluxes for Caribbean tropics are not well constrained, but our canopy average is comparable to one prior LF estimate of 29 µg m$^{-2}$ y$^{-1}$ for the surrounding Luquillo Experimental Forest (LEF)[48]. The substantial increase in tropical foliar fluxes when understory vegetation is included emphasizes the importance of tropical ecosystem complexity in driving high rates of Hg deposition[49]. Our data thus support the notion of tropical forests as hotspots of Hg deposition driven by vegetative uptake and high net primary productivity[4,48]. Similarly, tropical Oe and Oa horizon accumulation rates are by far highest among all measured soils, further supporting high Hg deposition as a key characteristic of low-latitude forests [$p < 0.05$].

## Hg accumulation and atmospheric flux in temperate forests

Another salient feature of soil Hg accumulation evident in Fig. 2 is a pronounced and significant increase in accumulation rates with soil depth [$p < 0.05$]. To contextualize this pattern, we compare soil Hg accumulation rates to independent atmospheric Hg flux estimates for each ecosystem type, first for temperate forests where total ecosystem Hg deposition data are available for the northeastern United States

(Fig. 2a); two sites, Harvard Forest (site HaF, deciduous) and Howland Forest (site HoF, coniferous), include micrometeorological GEM flux quantification[3,22], and a third Downer Forest (site DoF, mixed)[50,51]. The primary pathways of atmospheric Hg deposition to soil systems include LF originating largely from GEM uptake; deposition by rain and canopy wash-off originating largely from oxidized Hg$^{II}$ (throughfall or TF); and non-foliar deposition of GEM directly to both forest floor (O horizon soil) as well as to long-lived above-ground tissues such as mosses, lichen, and tree bark (Fig. 1)[5]. The non-foliar GEM component is calculated here as the above-canopy GEM flux minus the foliar component, and the latter is approximated as LF flux. Together these sum to total ecosystem flux (EF). Deposition studies at the three temperate sites thus yield the following flux estimates (mean ± SD): LF = $11.8 \pm 0.6$, TF = $7.8 \pm 0.9$, non-foliar GEM = $7.5 \pm 7.5$, and EF = $28.2 \pm 6.0$, µg m$^{-2}$ y$^{-1}$.

Hg accumulation rates for temperate forest soils are compared with measured LF, TF, and EF atmospheric fluxes in Fig. 3 for calendar years 2019–2022 (measured 1–5 years post soil collection). Both deciduous and coniferous soils follow a pattern of strongly increasing Hg accumulation rates with depth (Fig. 3a; we call this typical pattern "accumulating"; see discussion below). Hg accumulation in foliage with ages < 1–3 years is consistent with local litterfall flux (LF) measurement as noted above. Accumulation rates in Oi horizon [ages $2.5 \pm 3.3$ years, mean ± SE] are consistently higher than LF fluxes and show additional Hg accumulation in the forest floor [$18.0 \pm 1.8$ µg m$^{-2}$ y$^{-1}$]. Ongoing TF and non-foliar GEM deposition can increase Hg deposition to the forest floor following LF[22,52–54]. Oi horizon accumulation rates are considerably lower than total EF measured by flux-gradient at both the HaF and HoF sites, however, with a cumulative shortfall averaging 45 µg m$^{-2}$, or 36% of total deposition. This "missing Hg" could indicate an overestimation of EF at these sites, re-emission of Hg as GEM back to the atmosphere, or more likely as discussed below, preferential translocation of Hg from the forest floor deeper into mineral soils.

In contrast to a missing Hg sink in Oi horizons, Hg accumulation rates in Oe soil horizons converge with contemporary EF [mean age $7.4 \pm 2.8$ years, flux $26.4 \pm 2.2$ µg m$^{-2}$ y$^{-1}$]. Hg accumulation rates in Oa horizons are higher still, nearly twice the contemporary ecosystem atmospheric Hg flux in soils aged 20–30 years [$45.9 \pm 3.3$ µg m$^{-2}$ y$^{-1}$]. Hg accumulation rates continue to increase in underlying mineral soil and peak at depths corresponding to calendar years 1975–2000 (Fig. 4). At these depths about half of our temperate soils (6 of 11) record robust histories of atmospheric deposition (Supporting Information, Table S2, Figs. S1–S3). Here, accumulation rates average

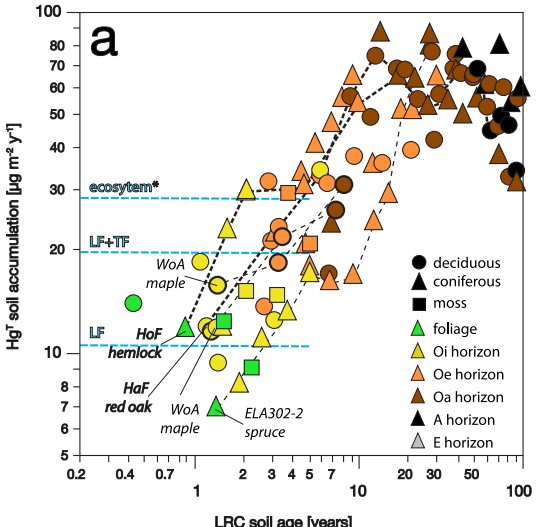
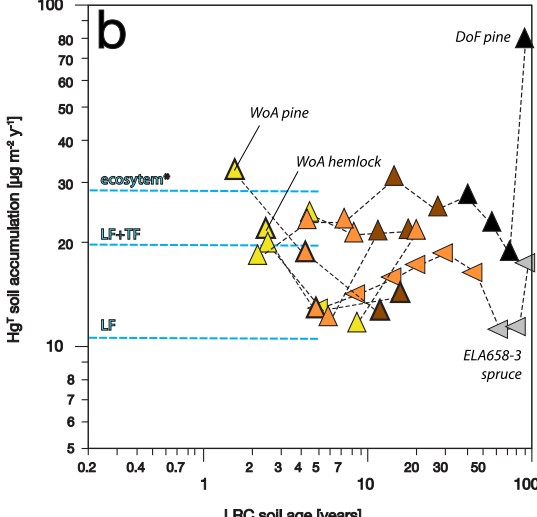

**Fig. 3 | Hg soil accumulation rate versus soil age in temperate/boreal forest soils. a** Typical Hg-accumulating soil. **b** Low-accumulating soils, all six are coniferous and one is boreal. *Contemporary atmospheric fluxes for the years 2019–2022 are shown as indicated, including litterfall (LF), throughfall (TF), and ecosystem flux, which is the sum of LF, TF, and non-foliar gaseous elemental mercury (GEM) deposition.

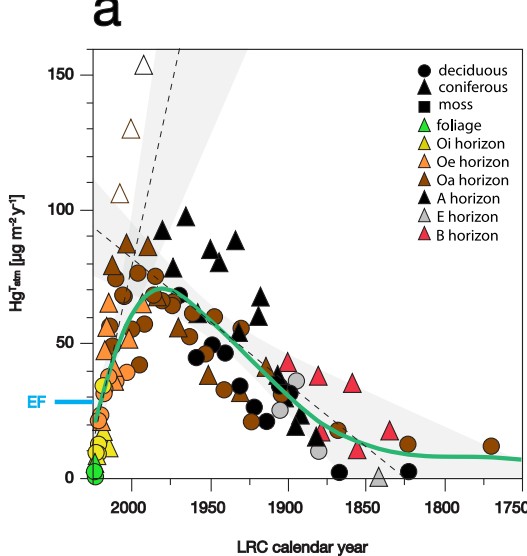
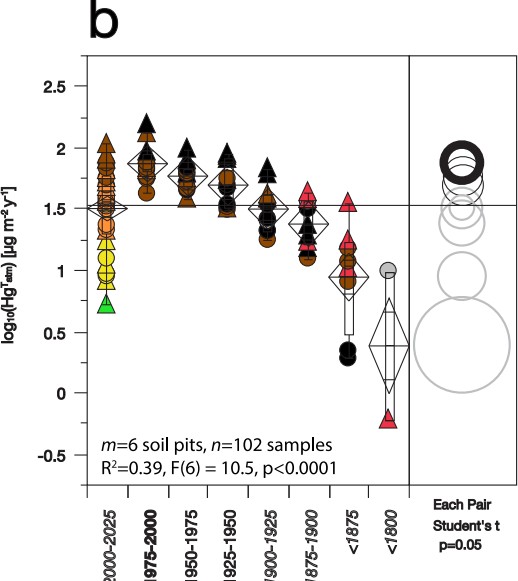

**Fig. 4 | Decadal history of atmospheric Hg accumulation in soil profiles of northeastern USA with robust chronometry. a** Reconstructed chronologies for soils ELA302-1, PiP01, PiP02, HaF01, HuB01, HuB02; see Supporting Information. Contemporary total ecosystem Hg flux (EF) measured by micrometeorology is indicated. Dashed lines show respective best fits for periods of increasing and decreasing rates of Hg accumulation, with shading for 95% confidence intervals. Solid green line shows spline fit. **b** ANOVA for Hg accumulation by period. Diamond plots show means ± 2SE. Box plots show medians and 25%, 50%, and 75% quartiles. Side panel illustrates Student's *t*-test, where a circle's diameter represents variance, and non-overlapping circles indicate significant differences in mean Hg fluxes. The 1975–2000 period (bold circle) has non-significant differences with periods 1925–1950, 1950–1975, and 2000–2025 (black circles). Remaining periods (gray circles) are significantly different [*p* < 0.05].

$73 \pm 5 \, \mu g \, m^{-2} \, y^{-1}$ which is ~2.5 times higher than measured contemporary atmospheric EF. Higher Hg accumulation in Oa and mineral soil horizons versus contemporary atmospheric fluxes suggests that some Hg may be preferentially translocated from overlying organic soil horizons. Most importantly, however, our measurements support evidence that historical Hg forest deposition fluxes in the northeastern United States have much been higher than at present[27,55,56].

The timing of peak soil Hg accumulation (1950–2000) is notably consistent with regional lake sediment archives, where deposition of atmospheric Hg peaked during 1970–1990 and has subsequently declined to present by about a factor of two[15,56–58]. If peak soil accumulation of $73 \pm 5 \, \mu g \, m^{-2} \, y^{-1}$ for the years 1975–2000 is representative of the total Hg EF in that era, it would be consistent with a similar decrease by a factor of 2 to contemporary terrestrial deposition for this region (Fig. 4). Hence, soil Hg reconstructions agree well with lake sediment records. Both archives are also consistent with measured declines in atmospheric GEM concentrations in North America that have averaged 1–2% per year since 1990[26,59].

The consistency among soil, lake, and atmospheric records provides strong evidence that, despite the complexity of pedogenic

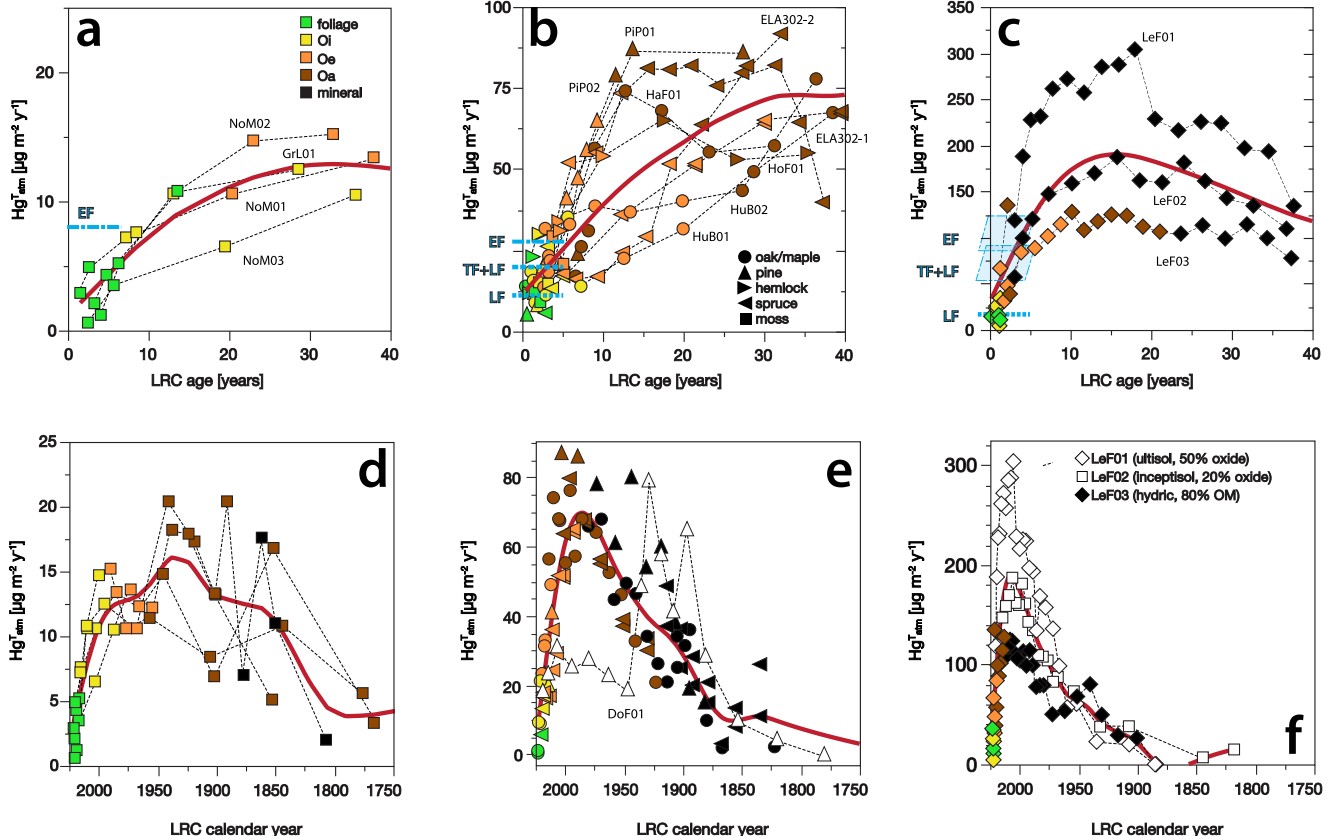

**Fig. 5 | Soil Hg accumulation rates in global soils across a latitudinal gradient.**
**a**, **d** Arctic tundra (AK, GR); **b**, **e** northeastern temperate (VT, NH, ME, MA) and boreal (ON) forests; **c**, **f** tropical forest (PR). Temperate data are reproduced from Fig. 2a. Soils highlighted in the discussion are colored white. Puerto Rico litterfall (LF), throughfall (TF), and ecosystem flux (EF) estimates are based on Shanley et al.[48,72].

processes and Hg biogeochemical dynamics, soil FRN chronometry provides direct constraints on historical terrestrial Hg fluxes. This opens unprecedented opportunities to assess historic Hg deposition across much larger spatial scales given the ubiquity and accessibility of soils. This line of reasoning further concludes that typical northeastern US soils are quantitatively efficient at retaining past atmospheric deposition of Hg from combined depositional sources, and that, while individual surface horizons are missing some fraction of ongoing Hg atmospheric deposition, this Hg is retained within the deeper soil profile. Our soil mass balance estimates hence indicate that substantial legacy emission of Hg from soils back to the atmosphere is unlikely across most soils (see discussion of low-accumulating soils below in the "Mercury low-accumulating soils and coniferous ecosystems" section), and instead suggest that typical temperate forest soils quantitatively retain atmospheric Hg ecosystem deposition over annual to decadal timescales.

### Latitudinal trends in forest soil Hg accumulation

Across contrasting forest ecosystems, reconstructed soil Hg accumulation rates demonstrate a strong latitudinal gradient in atmospheric Hg deposition, on the order of 10 µg m$^{-2}$ y$^{-1}$ for Arctic, 50 µg m$^{-2}$ y$^{-1}$ for temperate, and 150 µg m$^{-2}$ y$^{-1}$ for tropical Luquillo sites (Fig. 5a–c). Despite wide-ranging fluxes, however, similar temporal patterns in Hg accumulation spanning ecosystems suggest that the processes impacting soil Hg sequestration and overall mass balance are global. Specifically, like their temperate counterparts, both tropical and Arctic soils show Hg accumulation rates that fall short of respective contemporary deposition estimates in surface soil horizons, and that eclipse ecosystem Hg fluxes in deeper soils of multi-decadal ages (Fig. 5d–f). The convergence between soil accumulation and

atmospheric deposition rates occurs at corresponding soil ages of 5–10 years across highly variable soil orders spanning Gelisol, Inceptisol, Spodosol, and Ultisol. The magnitude of this effect also varies in accordance with ecosystem properties; for example, in the Luquillo tropical forests, soil Oi horizons accumulate Hg at just about 20% of estimated EF, versus 70% in temperate forests. We interpret this difference as a decreasing ability of surface organic soils to sequester Hg with higher mean annual temperatures and SOM mineralization rates. In Arctic samples, the Oi horizon Hg fluxes average 120% of contemporary EF flux, which is likely within reasonable uncertainty of estimated atmospheric deposition and indicates a strong retention of Hg by Arctic soils, as well[2]. We note, too, that the tundra Oi horizon is sufficiently old (up to 40 years) that it records higher rates of Hg deposition that predate successive emissions reductions implemented by the Clean Air Act (1970), Clean Air Act Amendments (1990), and Minimata Convention (2017).

Centennial histories of Hg accumulation in both Arctic and tropical soils mirror those of temperate forests and are similarly consistent with anthropogenic emissions of the global north, rising steeply through the twentieth century before declining around 1990–2000 in line with the implementation of international emissions controls (Fig. 5d–f). The precise magnitude and timing of peak soil accumulation may be influenced by variable SOM decomposition and colloid migration rates with soil depths among diverse soil orders (Fig. 5d–f). Shifts in peak Hg accumulation from that of mid-latitudes in Arctic (later) and tropical soils (earlier) might also reflect preferential affinity of Hg for humic matter in Oa horizons. Yet, within each ecosystem, the consistency of Hg accumulation patterns suggests that our Hg accumulation reconstructions accurately reflect the dynamics of atmospheric Hg deposition and accumulation across latitudes. For example,

we observe good correspondence in temporal trends among highly different tropical soils which include both hydric Inceptisols (site LeF03, e.g., 80% organic matter by weight) and Ultisols (site LeF01, e.g., 50% Al- and Fe-oxides by weight). Peak deposition in tropical soils appearing later (ca. 2000) than in temperate latitudes may also be influenced by the unique disturbance history of Puerto Rico. Here, Hurricane Maria (2017) caused massive defoliation of the Luquillo forest canopy that delivered 60% of above-ground forest biomass to the forest floor[60], along with the large pool of Hg contained therein. In Arctic soils, the period of peak Hg deposition is less clear due to large uncertainties in soil Hg reconstructions for years prior to 1950, which is a result of low atmospheric deposition of both FRNs and Hg, combined with slow SOM decomposition and colloid migration rates. Overall, the paucity of both FRN and Hg data for high- and low-latitude sites should be a focus for future work.

## Forest Hg dynamics and mechanisms of soil sequestration

All surface soil horizons (forest floor Oi horizons) show a substantial shortfall of Hg relative to atmospheric fluxes, at rates of 10 μg m$^{-2}$ y$^{-1}$ for temperate systems or even approaching 100 μg m$^{-2}$ y$^{-1}$ for tropical forests of Puerto Rico (Fig. 5). A large body of work supports the proposition that Hg in the forest floor is reduced to GEM, both during SOM decomposition and following wet deposition of Hg$^{II}$, and that this potentially yields a highly mobile GEM pool that could be subject to re-emission[6,14,18,19]. In contrast to the notion of significant net GEM emission, however, soil Hg accumulation rates increase in deeper soils far exceeding contemporary depositional fluxes, and as discussed above, in Oa to upper mineral horizons are quantitatively in agreement with historical changes from other independent deposition records. The mass balance thus requires that Hg lost from surface soils must be redistributed in a net downward direction that augments long-term sequestration.

We propose that percolation is likely key to facilitating mass transport of Hg from surface organic horizons, analogous to how FRNs reach soil depths of 5–10 cm or more during precipitation events (Fig. S4), and how TF is enriched in total Hg by factors of two or more over incident precipitation by washing Hg from foliar and canopy surfaces[5,61–64]. This is surprising since Hg is deposited to soils primarily through LF and direct deposition of GEM to the forest floor rather than by wet deposition. We suggest that Hg may be translocated as GEM preferentially to FRNs by rainfall and snowmelt during organic matter decomposition. To any extent that GEM may be re-emitted from surface horizons back to atmosphere, our mass balance suggests that this GEM is recycled back to soils, rather than lost to the atmosphere. Although enigmatic, such internal GEM recycling could occur via TF or LF of fine and coarse debris and may occur over decadal timescales. For example, on the order of 16% of Hg in TF may represent GEM that is re-deposited within the canopy from soil re-emission of prior Hg deposition[22]. At the same time, the stable isotope composition of GEM emitted from soils appears to be identical to that of atmospheric GEM and dissimilar from soil legacy Hg, which suggests that measured soil GEM re-emission reflects the transient turnover of atmospheric GEM rather than net emission from legacy soil reservoirs[19,65]. Direct re-emission of GEM from foliar surfaces may also be important for recycling wet-deposited Hg to the atmosphere before it has the opportunity to enter underlying soils[14], with an especially strong impact on the mass balance of isotope spike experiments such as METAALICUS. There, 50% of wet-deposited Hg was recycled back to the atmosphere from the forest canopy over a period of months[66].

In contrast to forest floor, underlying organic and mineral horizons are closed systems with respect to Hg. Pore-gas GEM concentrations there are typically lower than atmospheric levels, suggesting that in deep soils any available GEM becomes firmly bound or oxidized to particle-reactive Hg$^{II}$[14,21,24]. Hg thus seems to be translocated from the forest floor in gaseous form or deposited directly to

mineral soil by percolation and thereafter continues its inexorable downward migration as organometallic colloid. Critically, the retention of Hg in deep soils locks in its historical pattern of atmospheric deposition, like that of the $^{241}$Am nuclear bomb pulse (Table S2)[31,32]. As such, the fate of Hg in soils is also closely aligned with that of soil carbon insofar as long-term storage of both Hg and C in soils is contingent on their mobilization from organic to mineral soils and stabilization by complexation with secondary soil minerals including authigenic clays and Al, Fe-oxides that prevent subsequent reduction and emission[67–70].

## Mercury low-accumulating soils and coniferous ecosystems

In contrast to typical Hg-accumulating soils, a subset of northeastern temperate and boreal soils (20% of sites) shows decreasing Hg accumulation rates with soil depth, which we call "low-accumulating" soils (Fig. 3b). These soils are all coniferous, yet we stress that there is no generic difference between deciduous and coniferous soils across sites, and that ecotype alone is therefore insufficient to explain these patterns. We highlight results from the Woody Adams Conservation Forest in Vermont (site WoA) where deciduous maple and oak soils display typical Hg-accumulating behavior, yet adjacent pine and hemlock soils are low-accumulating. Although the pine and hemlock organic soils have total Hg loads that are higher per horizon than the hardwoods by factors of two to three, this is attributable largely to their older ages, slower decomposition rates, and greater accumulation of SOM. In terms of mass balance, however, the low-accumulating coniferous soils are "missing" 250 μg m$^{-2}$ or about 10% of accumulated Hg over the past 20 years relative to the deciduous sites. The nearby Downer Forest (DoF) coniferous pine soil, also low-accumulating and for which we have collected a full soil profile, shows a strong peak of Hg accumulation in the underlying mineral soil (Fig. 3b), suggesting that in some cases podzolization (downward mobilization of iron and aluminum oxides along with dissolved organic carbon) may preferentially mobilize Hg relative to the FRN tracers or bulk carbon. The coniferous 302-2 and HoF01 soils similarly show strong and atypic Hg enrichment in mineral horizons which we attribute to lateral podzolization along horizontal flow-paths (Fig. S2). While Hg may be decoupled from both FRNs and carbon in these cases and cannot be dated per se, here the FRNs provide a means for identifying the mass transfer of Hg even if quantifying its time of deposition and rate of redistribution will require additional new approaches.

More research is needed to confirm whether Hg in low-accumulating surface soils is lost to the atmosphere via re-emission, leached to mineral horizons, or exported from watershed soils by runoff. The Hg shortfall from individual surface soils approaches 20 μg m$^{-2}$ y$^{-1}$, which is comparable to typical contemporary atmospheric flux, and therefore would be large enough to be readily measured by atmospheric or watershed studies if such losses occurred at large scales. On the other hand, if the frequency of low-accumulating soils (20%) is representative of northeastern forests and such losses are limited to this minority of soils, the Hg ecosystem loss based on our mass balance would equate to only about 2% of annual EF. This scale is compatible with the re-emission of 0.9 μg m$^{-2}$ y$^{-1}$ measured by micrometeorology from the HoF coniferous Podsol forest floor[22]. At the watershed scale, isotope tracer experiments similarly suggest that about 1% of new deposition may be exported from upland forests in streamflow[9,71], although the export of cumulative legacy deposition may approach 10% of contemporary watershed input[72–74]. Weaker retention of Hg in coniferous forest organic soils and the possible influence of podzolization on mineral horizons described here demands further study as it may potentially be linked with high riverine export in boreal rivers and impact boreal and Arctic soil Hg retention under global warming[75], with implications for downstream food webs[76,77].

Typical uncertainties on soil Hg accumulation rates due to errors in the LRC age models are estimated to be on the order of 10% for soil ages <60 years[31]. High precision of the age models coupled with typically congruent behaviors between soil Hg and the FRN tracers allow us to reconstruct confident estimates of Hg deposition, peaking at $73 \pm 5\ \mu g\ m^{-2}\ y^{-1}$ in northeastern forests. Where Hg diverges from historical patterns of deposition, however, for example in low-accumulating soils, the concordance of multiple radionuclide tracers in the studied soil systems demonstrates that Hg mass loss can be confidently attributed to the specific physicochemical behavior of Hg. Given this, the considerable variation in Hg accumulation rates among northeastern soils is apparent in Figs. 3 and 4 reflects spatial variation in both how and where Hg is deposited and sequestered across the terrestrial landscape. Variable accumulation of Hg is underscored by a factor of five higher coefficient of variation among soil Hg inventories in comparison to $^{210}$Pb and $^{241}$Am (100% versus 20%; Table S1) and thus indicates the potential for high patchiness of Hg accumulation and hotspots of Hg losses.

## Insights into Hg biogeochemical cycling in soils

We demonstrate the direct estimate of soil Hg accumulation rates in upland soils using a "bottom–up" mass balance based on FRN chronometry, thereby establishing both reference inventories and accumulation rates for Arctic, temperate, boreal, and tropical soils impacted by global atmospheric Hg. Reconstructed rates support the magnitude of deposition measured in micrometeorological studies, which together affirm that GEM-derived atmospheric deposition is dominant across forest ecosystems. Our soil mass balance measurements show that surface soil horizons under-accumulate Hg while deeper organic and mineral horizons over-accumulate Hg compared to modern atmospheric deposition estimates. Superimposed on recent declines in rates of Hg atmospheric deposition, we propose that this is facilitated by vertical translocation of Hg, possibly mediated by GEM dynamics, percolation, and/or colloid transport, from surface to deeper soil horizons. In deeper organic and mineral soils, our data provide evidence of a closed system wherein Hg is firmly sequestered and thereafter regulated by colloidal transport at migration rates that are accurately measured by FRN chronometry. This application of FRN chronometry to terrestrial Hg dynamics thus establishes an exciting opportunity to expand conventional archives such as lake sediments, peat, or ice records to include soils as a largescale, widespread, and easily accessible medium to quantify Hg depositional processes and historical fluxes[28,29,75,78,79]. A key distinction is that soil accumulation rates accurately represent terrestrial deposition where inputs from vegetation uptake of GEM dominate (>70%) and result in terrestrial fluxes 4–5 times higher than measured in conventional archives. Flux reconstructions based on lake sediment records may underestimate total ecosystem Hg loads, in contrast, because they are biased to wet deposition, dependent on watershed connectivity, and are convoluted by impacts of land use that impact sedimentary Hg flux[80].

Reconstructed accumulation rates for contrasting ecosystems confirm enormous gradients in Hg deposition across global biomes previously reported in atmospheric measurements, ranging from $10\ \mu g\ m^{-2}\ y^{-1}$ in the Arctic to ~$30\ \mu g\ m^{-2}\ y^{-1}$ in northeastern US forests and over $100\ \mu g\ m^{-2}\ y^{-1}$ in tropical forests of Luquillo. Importantly, extraordinary rates of Hg accumulation observed in the "clean air" site of Puerto Rico (exceeding $300\ \mu g\ m^{-2}\ y^{-1}$ at one location) demonstrate that low-latitude tropical forests may be inherently strong Hg sinks due to a combination of high net primary productivity, complex disturbance histories, and high rainfall totals with access to a global pool of oxidized Hg$^{II}$ in the upper troposphere through deep convection[15,48]. Our measurements demonstrate that very high Hg fluxes in the tropics are not dependent on nearby pollution sources[63]. Few direct measures of atmospheric Hg deposition are available in tropical forests, however, and given their likely critical role in global Hg cycling, high priority should be placed on improving our understanding of Hg depositional pathways and accumulation in these critical ecosystems[17,25,48].

Our data contradict reports of widespread soil Hg losses by GEM re-emission and indicate that if such losses occur, they might efficiently re-cycle within ecosystems and back to soils rather than being exported to the global atmosphere. This might further imply that conventional LF and TF measurements overestimate net Hg fluxes by including a recycled component. Soil reconstructions across larger scales now have the potential to resolve this uncertainty in contemporary and historical Hg fluxes to terrestrial ecosystems, as well as to better understand the role of soil processes in Hg global mass balance. Atmospheric Hg concentrations are now slowly declining in North America, Europe, and East Asia due to emission controls, although emissions continue to increase in South America. Under declining Hg emission scenarios, the fate of legacy Hg stored in the soil becomes increasingly important, whether certain ecosystems such as boreal peatlands shift to become net sources that offset reductions in industrial emissions[13,14,16]. Our data strongly suggest that, with exceptions of surface litter layers and a subset of coniferous soils possibly linked to podzolization, most forest soils across latitudinal gradients are strong net sinks and act as a closed system with respect to atmospheric Hg.

The potential of FRN chronometry for predicting the fate of atmospheric Hg is twofold. First, in most soils, Hg behaves congruently with a suite of FRN radionuclide chronometers and this allows us to reconstruct robust decadal histories of Hg EF. It must be emphasized that critical physicochemical processes including reduction of Hg$^{2+}$ to GEM, podzolization, and possibly preference of Hg for thiol ligands[81–83], may fractionate Hg from FRN tracers in certain environments. In these cases, the second potential of FRN chronometry is an opportunity to quantify the extent to which Hg may deviate from historical expectation and thereby provide insight into drivers of Hg environmental behavior such as the mobilization of GEM from organic to mineral soil. This may be pursued with future work that couples Hg stable isotope dynamics with the FRN tracer $^{210}$Pb in organometallic colloid formation and transport, towards how these may act to sequester Hg in mineral horizons, or alternatively to preferentially export Hg through hydrologically connected watershed elements to discharge in aquatic and marine ecosystems, where it ultimately bioaccumulates in food webs to cause negative health outcomes for both wildlife and human populations.

## Methods

We measured Hg accumulation rates in foliage and soils using high-precision gamma spectrometry and high-resolution quantitative soil pits (Table S2). Our experimental design includes (1) contrasting canopy tree species within sites, (2) contrasting sites with identical species, and (3) contrasting ecosystems across climate and elevational gradients. Selected sites represent soil orders that include Inceptisol, Spodosol, Gellisol, and Ultisol. We focus on temperate forests in northeastern US states (Vermont, VT; New Hampshire, NH, Massachusetts, MA; and Maine, ME) and boreal forests in Ontario, Canada. Sites in Vermont include multiple forest types with the same lithology, elevation, and climate, with forest stands dominated by maple (*Acer saccharum*) and beech (*Fagus grandifolia*); red oak (*Quercus rubra*); white pine (*Pinus strobus*); or eastern hemlock (*Tsuga canadensis*). Sites in Massachusetts (*Quercus rubra*, Harvard Forest) and in Maine (*Tsuga canadensis* and *Pinus strobus*, Howland Forest) have published records of total Hg EF measured by micrometeorological flux-gradient methods in addition to TF and LF estimates[3,22]. The DoF site in Vermont has new multi-year records of Hg LF and TF that are reported here. One boreal site in Ontario, Canada, was also sampled where canopy is dominated by black spruce (*Picea mariana*).

Temperate and boreal sites were contrasted with tundra sites on the Seward Peninsula, Alaska (AK) dominated by moss (Sphagnum)[84] and Sisimuit, Greenland (GR) with shrub cover of black crowberry (Empetrum nigrum) over Sphagnum. Tropical sites include an elevational gradient (~350–1000 m a.s.l) in the LEF, Puerto Rico, that spans subtropical wet, lower montane, and rain forest life zones[85]. The LEF sites include distinct forest communities of Tabonuco (Dacroydes excelsa–Manilkara bidentata), Palo Colorado (Cyrilla racemiflora–Micropholis garciniifolia), and elfin woodland (Tabebuia rigida–Eugenia borinquensis), respectively[86,87].

Soils were collected from 30 × 30 cm or 50 × 50 cm quantitative pits, at hilltop, ridgeline, and locally flat locations selected to isolate vertical flowpaths[31]. For 16 short soil profiles consisting of uppermost organic soils (max. depth 5 cm), samples of thickness 0.5–1 cm were excavated by hand corresponding to Oi, Oe, and Oa organic horizons[32]. Eighteen additional long profiles in 1–5 cm increments to soil depths of 30–50 cm are also reported here. A total of 379 samples were analyzed and dated. Due to the large sample masses required for FRN analysis soils and foliage were oven-dried at 50 °C rather than by lyophilization, which previously has shown non-detectable to minor losses during drying (max. 4% of total Hg)[88]. Foliage and litter for Hg analysis were homogenized using a stainless-steel Wiley Mini-Mill (Thomas Scientific), and soils were disaggregated by agate mortar and pulverized in a zirconia ball mill.

Atmospheric Hg in soils was resolved from geogenic contributions using Al as a reference element[2,89], with total atmospheric Hg calculated as the sum of $[Hg_i – Al_i \times (Hg/Al)_{mineral}]$ for $i$ layers with ages <225 years, and the mineral component taken for soils older than the reference year 1800 which is the practical limit for reliable FRN chronometry. This procedure provides a conservative estimate of anthropogenic Hg accumulation in soils, although we recognize that earlier emissions occurred and could be significant near regional sources. For measurements of Al and major/trace elements, samples were digested with a mixed acid approach using $HNO_3$–$HCl$–$HF$–$H_2O_2$ and microwave heating with hot plate evaporation in Teflon vessels, and 150 mg samples/15 mL acid. Major/trace elements were measured by ICPOES (Spectro ARCOS) with reference materials NIST 1547 (peach leaves), NIST 2706 (New Jersey soil), NIST 2710 (Montana Soil II), USGS G3 (granite), CCRMP TILL 1 and 2 (glacial till), and STSD 1 and 2 (stream sediment). Recoveries for Al averaged 104 + 3%, 95 ± 2%, 104 ± 4%, and 97 ± 3%, 96 ± 3%, 99 ± 3%, 93 ± 2% and 98 ± 6% (mean ± SE), respectively.

Canopy foliar Hg inventories were estimated by multiplication of leaf Hg concentration by leaf area index and leaf mass area, or in one case by whole-tree harvest. Soil Hg inventories for each depth increment were estimated by multiplication of soil Hg concentrations by total recovered mass per sampling area of the quantitative pit (kg m$^{-2}$) rather than with an estimate of soil bulk density. Hg concentrations were measured by DMA (Milestone DMA-80) with reference materials NIST 2706 (New Jersey soil), and NIST 1547 (peach leaves), run every ten samples. Recoveries averaged 103.4 ± 4.8% (mean ± SD, $n = 98$) using standards of ~5 ng total Hg, and 99.3 ± 8.6% with standards of ~1 ng total Hg ($n = 98$), respectively.

Radionuclides were measured by high-precision gamma spectroscopy (Mirion BeGE 3830 gamma detectors)[90]. The FRN LRC model based on the $^7Be$:$^{210}Pb$ ratio was implemented as described by Landis et al.[31]. LRC soil ages were confirmed with the $^{241}Am$ bomb-pulse chronometer in deep soil pits (depths of 30–50 cm; Table S2)[31]. For shallow soil pits that included only organic horizons (FLH pits, Table S2), the $^{228}Th$:$^{228}Ra$ method was previously used to verify LRC ages in the 0–5-year age range[32]. (Table S2). We also report the more familiar bomb-pulse radionuclide $^{137}Cs$, but we stress that it is unreliable in soil systems due to the strong biological cycling of cesium as a potassium analog[31,91].

Statistical analyses were performed in JMP16 Pro, using analysis of variance (ANOVA) to compare Hg concentrations, inventories, and accumulation rates among contrasting soil horizons and across contrasting ecosystems.

## Data availability

Data generated in this study have been deposited in the Mendeley Data database at https://doi.org/10.17632/vs2mc22wft.1.

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

## Acknowledgements

This project was funded by the Department of Earth Sciences at Dartmouth College through the Fallout Radionuclide Analytics facility (FRNA). Visit us at sites.dartmouth.edu/frna/. Special thanks to Tatiana Barreto, Lee Hrenchuk, Ken Sandilands, Raquel Portes, Diego Spinola, and Craig Layne for invaluable contributions at field sites that made this work possible. We respect the sanctity of natural areas and the stewardship of their local communities. In this work, we have made all efforts to include local scientists and we acknowledge their contributions with authorship or acknowledgement where appropriate. All research activities were carried out under applicable review and permitting processes that protect natural and experimental areas, and all effort was made to minimize the impacts of our sampling activities. Samples from Luquillo Experimental Forest were obtained by permission of US Forest Service, El Yunque Special Use permit #YNF3019. Samples from Downer State Forest were obtained by permission of Vermont Department of Natural Resources, Special Use permit #22857. Foreign soils were imported under applicable USDA APHIS permits. Samples from Seward Peninsula, AK were obtained with funding to MP (NSF OPP-ANS-2116471). Samples from Harvard and Howland Forests were obtained with funding to DO (NSF EAR-1848212). Contributions from WM and TB were supported by NSF EAR 2012403 and NSF DEB 1831592.

## Author contributions

J.D.L. conceived and designed the project. J.D.L. and D.O. wrote the manuscript. J.D.L., V.F.T., J.Z., J.D.V. and F.M. collected samples in the field. J.D.L. and V.F.T. performed analytical and laboratory procedures. J.D.L., D.O., J.Z. and C.E.R. analyzed data. All authors contributed to manuscript edits. D.O., W.M., C.N. and M.P. provided material support that was essential to project success.

## Competing interests

The authors declare that financial or personal competing interests in the conduct of this study.
