## [Peer Review File · Nature Communications]

Quantifying soil accumulation of atmospheric mercury using fallout radionuclide chronometryREVIEWER COMMENTS

Reviewer #1 (Remarks to the Author):

This manuscript demonstrates a novel means by which to characterize the history of atmospheric deposition of Hg to soil through the application of independent fallout radionuclide chronometers. The results obtained by this study are significant to the field of Hg geochemistry in particular, and to the general issue of using soils as an archive of recent environmental change in the annual to decadal scale. The chronometry of Hg depositional fluxes obtained by this method are consistent with other published records, and the manuscript cites numerous published studies to validate this consistency. The work supports the claim that the fallout radionuclide approach can be used widely to investigate Hg deposition history across the globe. Although soils are complex and heterogeneous, the authors have taken a viable approach in sampling and analysis that deals well with the heterogeneity, and they used a concordance criterion (cumulative $^{241}\text{Am}/^{210}\text{Pb}$) to weed out disturbed soil profiles. The analytical methodology used by the authors is sound and sufficiently described to allow reproduction of the results by others. The first author is one of the leading experts in the application of fallout radionuclides to investigations of soil accumulation rates and the physical and chemical behavior of soils.

Overall the manuscript is well-written and deals well with uncertainties in the data. The presentation and statistical treatment of the data is of high quality. In this reviewer's opinion, this is an important manuscript that can be published nearly in its present form, with only minor revision in response to the points noted below. It is certainly timely, of broad interest, and suitable for publication in NATURE COMMUNICATIONS.

Specific comments on text:

- In the Introduction, the authors refer to the $^{228}\text{Th}/^{228}\text{Ra}$ chronometer, and the reader might assume that this was applied in the present study. However, these isotopes do not appear to have been measured as part of this study (no data for ^{228}Th or ^{228}Ra appear in Table S1). For clarity up front, the authors should indicate which measurements were made in this study (e.g. somewhere in the paragraph beginning at line 97).

- Table S1 includes measurements of the ^{137}Cs inventory in the soil profiles, but these data are not discussed in the manuscript. It would be worth mentioning ^{137}Cs in the Introduction (paragraph beginning at line 97), and also worth including a brief sentence or two discussion about what is learned from the ^{137}Cs inventory data somewhere later in the text or in the SI; if not, perhaps these data should be deleted from Table S1?

- The caption to Figure 3 needs some editing. The two panels are not labeled "a" and "b", even though there is reference to (b) in the caption. Also, in the right side of panel (b), there are different colored circles that indicate Students' t-test results of each pair, but it is unclear what the colors indicate or which pairs are being compared. A bit more explanation of this in the caption would be helpful for the

reader to understand this part of the figure.

- In the last sentence of the Figure 4 caption, “throughfall (LF)” should be corrected to “throughfall (TF)”.

Reviewer #2 (Remarks to the Author):

The manuscript entitled, “Quantifying soil accumulation of atmospheric mercury using fallout radionuclide chronometry” by Landis et al. uses a novel soil chronometer to estimate soil accumulation rates for soil of varying age. The complexity of soil processes and development have made elucidating terrestrial mercury cycling difficult to tease apart, a major problem given the importance of soil within the broader mercury cycle. Thus, the new chronometer holds great promise for the soil community broadly and the mercury community, specifically. The study specifically looks at how mercury accumulation rates vary across latitude, ecosystem type, soil horizon, and soil age. Several interesting patterns are documented through this approach, exemplifying the utility of the chronometer. Perhaps the two most salient findings are the dating of peak mercury accumulation to the 1950-1990 period and the implications from the mass- balance, which suggest relatively little soil mercury is lost to the atmosphere. It is the latter point which gives me pause, considering some of the possible limitations of the approach used here (expanded on below). Overall, I found the manuscript well written, timely, impactful, and generally quite interesting.

Major Comments:

1. The LRC model produces an estimate of average SOC age and the authors have provided convincing evidence of its ability to do so. Although this approach has advantages over other methods for soil dating (e.g., the authors point out limitations with ^{14}C), I would think there are still many of the same problems with connecting soil mercury processes with SOC age. In fact, the authors do an excellent job of highlighting some of these in their accompanying paper.

A) any Hg process independent of SOC could bias results. For instance, if the mineralization of older SOM was accompanied by a release of mercury and that mercury was subsequently bound by other (perhaps newer) SOM, the age model would no longer accurately reflect the deposition age of mercury.

B) the estimated age of the soil is really the average of a mixture of both young and new carbon and thus could be young and new mercury (which the authors point out in their companion paper, See Figure 2).

C) A number of physical processes may affect the movement of soil SOC and thus affect reservoir masses (and therefore accumulation rates). The accompanying paper presents several examples (lateral flow, vertical flow, etc.).

I think that addressing some of these limitations is necessary. Alternatively, highlighting that the other paper discusses them at length would be helpful. While I do not think they are particularly problematic for most of the data interpretation, I think they become an issue for the mass balance approach. The above three points could all result in biases of the actual accumulation at any given point in time. In fact, the isotopic mass balance in the accompanying paper makes this point quite clearly and, notably, that is with a single pulse (i.e., no previous deposition). Here, the mass balance reflects mercury deposition since soil formation.

In summary, I think the uncertainty in the mass balance should at the very least be commented on, given the importance of the finding. Perhaps you can report the ELA mass balance findings and some of the possible sources of variability. I think this point is especially important as readers may not be working through both documents concurrently.

2. In the bottom row of figure 4, the peak accumulation rates all roughly occur at the O_a/mineral interface, but the peak year gets younger as latitude decreases (~1950 for arctic, ~1970 for temperate, and ~2000 for tropical). It seems like these patterns could also be explained by the different turnover times of SOM, with the largest amount of mercury associated with the largest, slowest horizon (i.e., the O_a). Do the authors have thoughts on this?

General Comments:

I think the abstract art looks very nice.

I mention this under figure 1, but are the concentration values presented in text and in Figure 1 aluminum normalized/atm?

59 – 61: It might be nice to add specific citations to the high and low estimates, for the convenience of readers.

127: Consider adding measurements of variability to horizontal averages and in the reporting of subsequent values.

191: Additionally, could the lower accumulation rate in litter reflect the fact that not all deposition is being taken up by the O_i horizon? If direct deposition or throughfall passed largely through the O_i to, say, the O_e, then there wouldn't be missing mercury.

328: Assuming that the DOF peak in the deep mineral horizon was due to increased vertical migration of mercury, wouldn't the carbon signature associated with it still be younger than the observed ~100 years? The "missing mercury" was deposited in the last 20 years, no? I may have misunderstood.

Figures

Figure 1 Comments:

Is mercury accumulation on a per cm basis or are you scaling up the inventory by horizon depth prior the accumulation calculation?

Are the mercury values in panel B total values or do they represent atmospheric values (i.e., Al normalized)? If it is the latter, I think it would be good to somehow denote that in the figure.

Figure 3:

This plot has a "(b)" but no "(a)" and no demarcation on the plots.

Does the left-most (a) plot consist of both organic and mineral soils? Could these be distinguished somehow?

I also would be interested in seeing a plot (a) with the concentration values. Perhaps as an SI inclusion. If my interpretation is correct, the right-most plot uses red for two different purposes: highlighting the "increasing accumulation" zone and for the different comparison circles. I would suggest using two different color schemes to improve clarity.

Line edits:

16: This sentence is worded somewhat awkwardly, and I would suggest reworking it. "Accumulated" alone reads odd. Additionally, this sentence neglects on-going emissions of mercury and direct releases

to soil.

17: I might specify that “typical soils” refers here to “typical forested and arctic soils”, unless you feel that the forested soils here are representative of other soil types (e.g., grasslands). The majority of US soils are not forested or arctic soils.

28: This sentence has four clauses, which make it a bit cumbersome to read and may lead to confusion. Consider reworking.

65: Consider cutting “...to soils”.

183: Should this read “...1 to 5 years post soil collection” instead? 470: “every” instead of “each”

Reviewer #3 (Remarks to the Author):

Key Results

This study utilizes novel FRN chronometry to compare Hg accumulation in upland soil horizons at different latitudes encompassing 3 distinct ecosystems: arctic, temperate, and tropical.

The study reports the highest Hg accumulation rates in tropical ecosystems, consistent with higher atmospheric deposition in the tropics. Considering the small number of tropical and arctic sites, these results need further verification, but they are indeed the first of their kind reported.

The study also reports increasing Hg accumulation with soil depth, in general. The authors note that foliage flux is lower than total ecosystem flux (EF), but that Hg accumulation in deeper soil layers exceeds EF, except in a few temperate soils dominated by conifers. Temperate soils cores make up the majority of the samples and it is noteworthy that exceptions arise here. Arctic and tropical have $n = 3-4$, which is concerning. Further studies could confirm or deny these results.

Validity

The authors do an excellent job describing how their data fits with previous work in lake sediments, atmospheric deposition, recycling of GEM, implications for atmospheric Hg mass balances, etc. I have some concerns about migration of Pb and other isotopes and how that impacts dating of the cores that may need to be addressed.

Significance

The work described in this manuscript is highly significant. Following up on this work will lead to better constraints on Hg mass balances in the atmosphere and a better understanding of how ecosystems may respond to decreases in Hg deposition. The work is certainly novel. Relying on 3 or 4 cores from the arctic and tropical ecosystems is the primary troubling factor. However, even without the comparisons across ecosystems the results for the temperate soils are significant in the field of Hg research.

Data and methodology

The authors do an excellent job framing the findings and implications of the soil cores, particularly

relative to reported atmospheric fluxes of Hg. Questions that I had after examining the figures were addressed clearly and impressively in the article. The examination of EF relative to the accumulation rates in the soil cores was quantitative and thorough.

The finding of some cores being low-accumulating is difficult to explain and ties into concerns regarding the dating of these soils. The authors offer plausible explanations. However, if there is significant downward transport of Hg via dissolved organic matter (DOM), which is offered as potentially explaining higher Hg accumulation in deeper soil horizons, how can the dating still be valid? Would not the ^{210}Pb also be transported downward and impact the dating? Also, if the Pb and Hg (and C) did NOT behave the same with colloidal transport, wouldn't this invalidate the dating of the Hg accumulation?

And furthermore, if there is downward movement of Hg there would also be lateral movement. Is this lateral movement insignificant, i.e. how much dissolved organic matter is transported out of each watershed? If DOM is being transported out there will be Hg associated with it. Is this transport negligible at the sites, especially those low-accumulating sites?

Analytical Approach

The study is very strong in this regard. More cores would have been preferred, but it is not very practical to get a large value of n for this type of field work.

Suggested improvements

The authors should address the limitations of the dating methods and the implications of DOM transport of Pb and other isotopes on the dating methods.

Clarity and context

This manuscript was exceptionally well-written and clearly states the implications of using this new method of dating soil cores. I really enjoyed reading this article and am excited by the results presented.

References

Appropriate

My expertise

Lake and peat coring are areas of expertise for me. I have always used ^{210}Pb dating and know some shortcomings of that method. The newer methods described here are not as familiar to me.

The Reviewer comments are reproduced in green text, and our responses are in black. Unless specified otherwise, line references refer to the revised manuscript.

With respect to the Editors comments requiring attention to limitations and uncertainties in the mass balance model, these are addressed throughout the revised manuscript as detailed below in responses to Reviewer comments.

Reviewer #1 (Remarks to the Author):

This manuscript demonstrates a novel means by which to characterize the history of atmospheric deposition of Hg to soil through the application of independent fallout radionuclide chronometers. The results obtained by this study are significant to the field of Hg geochemistry in particular, and to the general issue of using soils as an archive of recent environmental change in the annual to decadal scale. The chronometry of Hg depositional fluxes obtained by this method are consistent with other published records, and the manuscript cites numerous published studies to validate this consistency. The work supports the claim that the fallout radionuclide approach can be used widely to investigate Hg deposition history across the globe. Although soils are complex and heterogeneous, the authors have taken a viable approach in sampling and analysis that deals well with the heterogeneity, and they used a concordance criterion (cumulative $^{241}\text{Am}/^{210}\text{Pb}$) to weed out disturbed soil profiles. The analytical methodology used by the authors is sound and sufficiently described to allow reproduction of the results by others. The first author is one of the leading experts in the application of fallout radionuclides to investigations of soil accumulation rates and the physical and chemical behavior of soils.

Overall the manuscript is well-written and deals well with uncertainties in the data. The presentation and statistical treatment of the data is of high quality. In this reviewer's opinion, this is an important manuscript that can be published nearly in its present form, with only minor revision in response to the points noted below. It is certainly timely, of broad interest, and suitable for publication in NATURE COMMUNICATIONS.

Specific comments on text:

- In the Introduction, the authors refer to the $^{228}\text{Th}/^{228}\text{Ra}$ chronometer, and the reader might assume that this was applied in the present study. However, these isotopes do not appear to have been measured as part of this study (no data for ^{228}Th or ^{228}Ra appear in Table S1). For clarity up front, the authors should indicate which measurements were made in this study (e.g. somewhere in the paragraph beginning at line 97).

Apologies for misleading. The $^{228}\text{Th}:^{228}\text{Ra}$ data were reported in a previous publication, but for many of the same samples now reported here for Hg (the "FLH" code we use in Table S1). We feel it is important to highlight the $^{228}\text{Th}:^{228}\text{Ra}$ data because they give extra confidence in the Hg fluxes we report, since the dating was done by multiple concordant methods. We have added text to clarify this at line 97 and referenced at line 509.

- Table S1 includes measurements of the ^{137}Cs inventory in the soil profiles, but these data are not discussed in the manuscript. It would be worth mentioning ^{137}Cs in the Introduction (paragraph beginning at line 97), and also worth including a brief sentence or two discussion about what is learned from the ^{137}Cs inventory data somewhere later in the text or in the SI; if not, perhaps these data should be deleted from Table S1?

We believe the ^{137}Cs data are important to include because readers familiar with fallout radionuclide dating in other systems (peat, sediment) will expect this corroborating bomb-pulse marker. In soils we use ^{241}Am because Cs is strongly biologically cycled and thus unreliable. We have included a new reference to this effect in line 510 (Methods) where we have introduced discussion of ^{137}Cs (Kaste et al. 2021 in our reference list). We chose not to discuss at line 97 in order to maintain readability of the Introduction, since there we are describing more broadly the advances in FRN chronometry that have made possible the present study of Hg.

- The caption to Figure 3 needs some editing. The two panels are not labeled “a” and “b”, even though there is reference to (b) in the caption. Also, in the right side of panel (b), there are different colored circles that indicate Students’ t-test results of each pair, but it is unclear what the colors indicate or which pairs are being compared. A bit more explanation of this in the caption would be helpful for the reader to understand this part of the figure.

We have labeled panels. We have changed colors of the circles for better clarity, and now include a description in the caption.

- In the last sentence of the Figure 4 caption, “throughfall (LF)” should be corrected to “throughfall (TF)”.

So changed, thank you for the careful read!

Reviewer #2 (Remarks to the Author):

The manuscript entitled, “Quantifying soil accumulation of atmospheric mercury using fallout radionuclide chronometry” by Landis et al. uses a novel soil chronometer to estimate soil accumulation rates for soil of varying age. The complexity of soil processes and development have made elucidating terrestrial mercury cycling difficult to tease apart, a major problem given the importance of soil within the broader mercury cycle. Thus, the new chronometer holds great promise for the soil community broadly and the mercury community, specifically. The study specifically looks at how mercury accumulation rates vary across latitude, ecosystem type, soil horizon, and soil age. Several interesting patterns are documented through this approach, exemplifying the utility of the chronometer. Perhaps the two most salient findings are the dating of peak mercury accumulation to the 1950-1990 period and the implications from the mass- balance, which suggest relatively little soil mercury is lost to the atmosphere. It is the latter point which gives me pause, considering some of the possible limitations of the approach used here (expanded on below). Overall, I found the manuscript well written, timely, impactful, and generally quite interesting.

Major Comments:

1. The LRC model produces an estimate of average SOC age and the authors have provided convincing evidence of its ability to do so. Although this approach has advantages over other methods for soil dating (e.g., the authors point out limitations with ^{14}C), I would think there are still many of the same problems with connecting soil mercury processes with SOC age. In fact, the authors do an excellent job of highlighting some of these in their accompanying paper.

We clarified (line 118) that a critical difference is that radionuclides accurately date exposure age of SOM whereas ^{14}C is heavily biased to include carbon recycling time. We agree that there is still a conceptual problem of dating SOM versus dating the Hg itself (more below).

A) any Hg process independent of SOC could bias results. For instance, if the mineralization of older SOM was accompanied by a release of mercury and that mercury was subsequently bound by other (perhaps newer) SOM, the age model would no longer accurately reflect the deposition age of mercury.

We agree and clarified (lines 487-493, 512-514, 590-598) that it is likely that Hg is sometimes decoupled and convoluted with SOM in ways not fully understood. However, while we are dating organic matter, the age model itself is based on (primarily) ^{210}Pb , and the central thesis of the manuscript becomes, whether Hg and Pb follow the same pathways during SOC decomposition. Our central hypothesis (lines 111-115) is that the FRNs (^{210}Pb) provide a normalizing term that isolates Hg flux from various biogeochemical transformations, and that Hg and FRNs should follow the same pathways during decomposition, *absent the influence of Hg reduction to GEM and possible subsequent mobilization*. Where Hg quantitatively follows Pb, we can truly date it (this appears to be the case in most soils studied given their converging accumulation histories). Where it does not, the model provides a way of identifying rates of losses (Hg mobilization).

B) the estimated age of the soil is really the average of a mixture of both young and new carbon and thus could be young and new mercury (which the authors point out in their companion paper, See Figure 2).

We agree, this is certainly true. The FRN ages represent averages as well (same as in sediment/peat records). To the extent that Pb traces Hg behavior, the LRC age represents average age of the deposited Hg. It is only when Hg behavior diverges from the tracers that the Hg "age" is substantially different than that of SOC to which it is bound. We added a short statement (line 650) to clarify this.

We also note that we referred to Hg deviation from tracers due to GEM "mobilization", "translocation", or "percolation" throughout the original manuscript at lines 97, 204, 263, 280, 306, 348, 362, 369, and 385. We believe these are calling out the very processes that Reviewer has cited, although perhaps not as explicitly as possible describing how these processes relate to the age model and mass balance. We have added text to emphasize that Hg can deviate from expectation based on FRNs, at line 417. We have also taken this opportunity to mention that Hg has ligand preferences that may also contribute to its fractionation from FRNs (now citing Jiskra et al. 2014, Chen et al. 2016, Zhang et al. 2018).

C) A number of physical processes may affect the movement of soil SOC and thus affect reservoir masses (and therefore accumulation rates). The accompanying paper presents several examples (lateral flow, vertical flow, etc.).

I think that addressing some of these limitations is necessary. Alternatively, highlighting that the other paper discusses them at length would be helpful. While I do not think they are particularly problematic for most of the data interpretation, I think they become an issue for the mass balance approach. The above three points could all result in biases of the actual accumulation at any given point in time. In fact, the isotopic mass balance in the accompanying paper makes this point quite clearly and, notably, that is with a single pulse (i.e., no previous deposition). Here, the mass balance reflects mercury deposition since soil formation.

The Reviewer has done an excellent job of summarizing the conceptual challenge to interpreting soil chronometry to Hg. The key to the method is appreciating that the LRC model, by way of dating SOC, provides a hypothesis for behavior of Hg. This is our key statement, at line 93. We have restructured this paragraph to emphasize this as it should guide the reader through the paper. Unfortunately, our companion paper is still in Review and we believe best to publish the present manuscript as quickly as possible, but note that the changes we have made, listed above, will make this clear to the reader.

In summary, I think the uncertainty in the mass balance should at the very least be commented on, given the importance of the finding. Perhaps you can report the ELA mass balance findings and some of the possible sources of variability. I think this point is especially important as readers may not be working through both documents concurrently.

We agree, per comments above. We also cite references to METAALICUS at line 460.

2. In the bottom row of figure 4, the peak accumulation rates all roughly occur at the Oa/mineral interface, but the peak year gets younger as latitude decreases (~1950 for arctic, ~1970 for temperate, and ~2000 for tropical). It seems like these patterns could also be explained by the different turnover times of SOM, with the largest amount of mercury associated with the largest, slowest horizon (i.e., the Oa). Do the authors have thoughts on this?

It is a good observation. We had already suggested that the SOM decay might regulate the extent to which Oi/Oe horizon retain Hg, with higher rates of mobilization in tropical soils. However, we added a sentence to refer to this possibility (line 422), i.e., that if the shift in apparent peak accumulation across biomes is related to SOM dynamics and preference for Oa humic matter, this would require that Hg migrate more quickly *downward* than Pb in the arctic soils, but more *slowly* in the tropics. It is unclear why this should be. We also emphasize that the tropical soils all show the same trend, despite LeF3 being 80% organic matter and the others 50% oxyhydroxide, about as different as soils can get! It is unclear why Hg would behave identically in these disparate tropical soils, and yet differ from temperate or arctic soils. Substantially more work will be required to resolve this, beyond present scope.

For the arctic soils we suggested that the late peak is very likely due to uncertainties in both age model and atmospheric Hg correction (lines 420-425) since it is very difficult to resolve both atmospheric ^{210}Pb and Hg from their geogenic counterparts. So, these combined, we believe more likely that Hg and Pb are still following similar behaviors across biomes, but that soils of each biome soil have specific challenges to the age dating itself that requires further work, and the histories in both Arctic and tropics require further study.

General Comments:

I think the abstract art looks very nice.

Thank you!

I mention this under figure 1, but are the concentration values presented in text and in Figure 1 aluminum normalized/atm?

Good catch. They are not, which we believe is important for comparability with other studies. In our data repository we include total Hg and total atmospheric Hg.

59 – 61: It might be nice to add specific citations to the high and low estimates, for the convenience of readers.

These were added at the sentence following the estimates but we have moved forward for clarity (Smith-Downey et al. 2020, Zhu et al. 2016, Zhou et al. 2020, Zhang et al. 2023).

127: Consider adding measurements of variability to horizontal averages and in the reporting of subsequent values.

We have omitted them since the averages are informational to provide a frame of reference, but variance would reflect cross-ecosystem range which is less straightforward to interpret. These are now added in a new Supporting Table.

191: Additionally, could the lower accumulation rate in litter reflect the fact that not all deposition is being taken up by the Oi horizon? If direct deposition or throughfall passed largely through the Oi to, say, the Oe, then there wouldn't be missing mercury.

Yes, we think it just so, and we clarified this in lines 430-436. This is the reason we discuss percolation as a dominant process for translocating Hg (line 308 and elsewhere) and described the depth distribution of ^7Be (Supporting Information, Figure S4) since this short-lived isotope (half-life 54 days) records primarily percolation and does not survive long enough to record subsequent advection/diffusion. This is surprising for

Hg, however, since most Hg enters soil as GEM either with litterfall or directly to forest floor, rather than by rainfall as for ^7Be . In this sense the Hg is missing from surface Organic horizons where we should expect it based on both foliar and non-foliar GEM deposition, but found deeper in the soil profile than we expect if GEM were to remain bound to this bulk organic matter.

328: Assuming that the DOF peak in the deep mineral horizon was due to increased vertical migration of mercury, wouldn't the carbon signature associated with it still be younger than the observed ~ 100 years? The "missing mercury" was deposited in the last 20 years, no? I may have misunderstood.

We would not expect the carbon age to be different than expectation, since in this case we expect that the Hg has moved deeper relative to either the FRN tracers or bulk carbon itself that the FRNs are dating. We believe more likely that Hg is mobilized by GEM or instead some highly mobile Hg-DOC complex, but in this case, yes, the apparent age of the Hg would be much older than reality. We have clarified at line 341 that the process would require *preferential* mobilization of Hg. We also caution over-interpretation of individual soil profiles since their local histories are difficult to constrain with certainty.

Figures

Figure 1 Comments:

Is mercury accumulation on a per cm basis or are you scaling up the inventory by horizon depth prior the accumulation calculation?

The figure shows Hg inventories on a cm^{-1} basis for panel C. For the accumulation calculation, the inventories [$\mu\text{g m}^{-2}$] are instead normalized by the time duration of each cm interval, which is actually one of the most powerful uses of the chronometry since depth is an arbitrary axis with respect to atmospheric fluxes. This is what allows use to reconstruct high-resolution histories, but it does inflate the variance of flux values we observe within an individual soil horizon due to changing fluxes through depth/time. We have added clarification to the figure caption.

Are the mercury values in panel B total values or do they represent atmospheric values (i.e., AI normalized)? If it is the latter, I think it would be good to somehow denote that in the figure.

These are not normalized to better compare with literature data. This is now clarified in the figure and legend.

Figure 3:

This plot has a "(b)" but no "(a)" and no demarcation on the plots.

Does the left-most (a) plot consist of both organic and mineral soils? Could these be distinguished somehow?

We now use the same shape and color scheme as previous figures to show both ecotype and soil horizon.

I also would be interested in seeing a plot (a) with the concentration values. Perhaps as an SI inclusion.

We have added this as requested. It is reassuring to see the same trends in both concentrations and fluxes, this is due to the preponderance of atmospheric Hg residing in organic soils with low geogenic contributions, and little in mineral soil where bulk density is much higher.

If my interpretation is correct, the right-most plot uses red for two different purposes: highlighting the "increasing accumulation" zone and for the different comparison circles. I would suggest using two different color schemes to improve clarity.

We now use the same shape and color scheme as previous figures and show both ecotype and soil horizon.

Line edits:

16: This sentence is worded somewhat awkwardly, and I would suggest reworking it. "Accumulated" alone reads odd. Additionally, this sentence neglects on-going emissions of mercury and direct releases to soil.

We have reworded, thank you.

17: I might specify that “typical soils” refers here to “typical forested and arctic soils”, unless you feel that the forested soils here are representative of other soil types (e.g., grasslands). The majority of US soils are not forested or arctic soils.

We have changed to 'forest and tundra soils'. Grasslands should be a priority for future work.

28: This sentence has four clauses, which make it a bit cumbersome to read and may lead to confusion. Consider reworking.

We changed this accordingly.

65: Consider cutting “...to soils”.

We changed this accordingly.

183: Should this read “...1 to 5 years post soil collection” instead?

Yes, changed accordingly, thank you!

470: “every” instead of “each”

So changed.

Reviewer #3 (Remarks to the Author):

Key Results

This study utilizes novel FRN chronometry to compare Hg accumulation in upland soil horizons at different latitudes encompassing 3 distinct ecosystems: arctic, temperate, and tropical.

The study reports the highest Hg accumulation rates in tropical ecosystems, consistent with higher atmospheric deposition in the tropics. Considering the small number of tropical and arctic sites, these results need further verification, but they are indeed the first of their kind reported.

We agree that applying our method to additional sites should be a high priority. The excellent consistency of the results at the two arctic sites (Alaska and Greenland), and within the one tropical site (Puerto Rico) gives good confidence that the method is reliable, but we need broader spatial coverage to understand global Hg dynamics.

The study also reports increasing Hg accumulation with soil depth, in general. The authors note that foliage flux is lower than total ecosystem flux (EF), but that Hg accumulation in deeper soil layers exceeds EF, except in a few temperate soils dominated by conifers. Temperate soils cores make up the majority of the samples and it is noteworthy that exceptions arise here. Arctic and tropical have $n = 3-4$, which is concerning. Further studies could confirm or deny these results.

We have added a comment about the low replication and pointed out the need for more sites (lines 285-286). However, we point out that the soils within the Arctic (both Greenland and Alaska) and tropics (only Puerto Rico) are very consistent internally. This provides a very strong latitudinal gradient that is of great interest and should be explored further.

Validity

The authors do an excellent job describing how their data fits with previous work in lake sediments, atmospheric deposition, recycling of GEM, implications for atmospheric Hg mass balances, etc. I have some concerns about migration of Pb and other isotopes and how that impacts dating of the cores that may need

to be addressed.

The corroboration of multiple age models including 5 different metals as well as ^{14}C gives us confidence that, to the first order and to such an extent to be an extremely useful method, we are measuring fundamental soil processes that very similarly impact a range of metals and carbon. That Hg is an outlier does in fact show that, in some fashion, it is preferentially mobilized, but that is of course the crux of our paper... that we can use the age models to reconstruct Hg fluxes in some cases, and in others to highlight very important aspects of Hg environmental chemistry. We have added clarity to this point in responses to Reviewer #2 above.

Significance

The work described in this manuscript is highly significant. Following up on this work will lead to better constraints on Hg mass balances in the atmosphere and a better understanding of how ecosystems may respond to decreases in Hg deposition. The work is certainly novel. Relying on 3 or 4 cores from the arctic and tropical ecosystems is the primary troubling factor. However, even without the comparisons across ecosystems the results for the temperate soils are significant in the field of Hg research.

We agree that temperate soils are the focus, and the arctic and tropical data, while limited, provide a strong gradient and important global perspective that at once is consistent with available deposition data for these regions, but also highlights the need for new work in these locations.

Data and methodology

The authors do an excellent job framing the findings and implications of the soil cores, particularly relative to reported atmospheric fluxes of Hg. Questions that I had after examining the figures were addressed clearly and impressively in the article. The examination of EF relative to the accumulation rates in the soil cores was quantitative and thorough.

The finding of some cores being low-accumulating is difficult to explain and ties into concerns regarding the dating of these soils. The authors offer plausible explanations. However, if there is significant downward transport of Hg via dissolved organic matter (DOM), which is offered as potentially explaining higher Hg accumulation in deeper soil horizons, how can the dating still be valid? Would not the ^{210}Pb also be transported downward and impact the dating? Also, if the Pb and Hg (and C) did NOT behave the same with colloidal transport, wouldn't this invalidate the dating of the Hg accumulation?

The fundamental challenge to using FRN for soil chronometer is understanding what it is that we are dating. Soils are generally not aggradational in the sense of sediment or peat, except perhaps in deep O horizons of cold climates, so the advection that we measure must be interpreted differently. The model demonstrates that ^{210}Pb migrates through soil at rates of a one to few mm y^{-1} , which is an order of magnitude higher than soil production, so must represent some DOM or colloidal process (described more in Landis et al. 2016). However, the concordance of multiple chronometers including ^{14}C that we discuss in lines 80-90 is critical for appreciating that this migration process incorporates both particle-reactive metals and carbon itself. The correlation between ^{210}Pb and ^{14}C first described by Dorr and Munich (1989) and now in a coming manuscript of ours for which data was provided in our Cover Letter, confirm that many metals and carbon are migrating together. Thus, the migration of ^{210}Pb does not invalidate the method, it is in fact the basis of the method, insofar as ^{210}Pb provides a tracer of this fundamental soil process(es).

The real question is, we believe, whether Hg and Pb will behave the same way. This is our critical hypothesis, as posed in lines 93-96: we hypothesize that Hg and Pb will behave the same, since C, Pb, Be, Ra, and Th tracers appear to behave the same insofar as their age models are concordant. If, however, there is evidence Hg does not, this becomes the point of acute interest and the secondary value of our method (the first being flux reconstruction). Our ability to see that in some coniferous forests Hg appears to be preferentially

mobilized is a key finding that points to a critical need for more work in these environments. We address these same points following the recommendations of Reviewer 2 above.

And furthermore, if there is downward movement of Hg there would also be lateral movement. Is this lateral movement insignificant, i.e. how much dissolved organic matter is transported out of each watershed? If DOM is being transported out there will be Hg associated with it. Is this transport negligible at the sites, especially those low-accumulating sites?

We agree that this is true. Our soils are collected in suitable reference sites where to every extent possible we try to isolate atmospheric deposition and minimize potential for lateral transport by sampling at hilltops and ridgelines in locally flat locations (now clarified in line 532 and more in Supporting Information).

Nonetheless, we cannot rule it out, and this is now specified at lines 342-348 and shown in Figure S2. We also cite rates of Hg export that may be expected in association with DOM and how these compare with our Hg estimates at lines 356-363.

Lateral subsurface flow is most likely the cause of high subsurface concentrations in soils ELA302-2 (bedrock depression, confined) and HoF (irregular till and boulder confined). But we do have some tools such as concordance of the LRC and ²⁴¹Am models and Am/Pb inventories and ratios (Figure S1), and now, expected Hg deposition inventories, to help rule out these situations. Truly it is extraordinary the degree to which these age models agree in dating the bomb-pulse horizon (better than 10%!).

Rather than a weakness of the dating, as we have argued to Reviewer 2, this really is a key importance of it. In localities where Hg inventories and fluxes do not conform to expectation based on histories of Hg deposition and regional soil inventories (smaller or larger), we can gain insight into processes that may be responsible. This is the approach that highlights the need for better understanding of Hg mobilization in coniferous forests since they are likely at greater risk of Hg accumulation in food webs (e.g., lines 355-358 and lines 367-369, 426-428).

Analytical Approach

The study is very strong in this regard. More cores would have been preferred, but it is not very practical to get a large value of n for this type of field work.

The Reviewer's understanding is much appreciated. The analytical demands for soil dating are extreme and there simply is not enough instrumentation available (anywhere). We have, to our knowledge, the largest academic facility in the US in terms of the number of appropriate state-of-the-art gamma detectors (large, Broad-Energy style), and we still must count each sample for 4 days to provide reasonable uncertainties for the FRNs. A single soil pit requires over one month of 24/7 lab time! We are trying very hard to double our capacity, but this requires about \$1.5M. Hopefully this publication will help sway the funding agencies!

Suggested improvements

The authors should address the limitations of the dating methods and the implications of DOM transport of Pb and other isotopes on the dating methods.

We have addressed this per comments to Reviewer #2 above. We note too that we have described the influence of podzolization (DOM transport) at multiple places throughout the manuscript.

Clarity and context

This manuscript was exceptionally well-written and clearly states the implications of using this new method of dating soil cores. I really enjoyed reading this article and am excited by the results presented.

Thank you for the careful read and the encouragement!

References
Appropriate

My expertise

Lake and peat coring are areas of expertise for me. I have always used ^{210}Pb dating and know some shortcomings of that method. The newer methods described here are not as familiar to me.

REVIEWERS' COMMENTS

Reviewer #1 (Remarks to the Author):

This review is of the revised version of NCOMMS-24-07443A. The authors have done a good job of responding to my previous review comments and suggestions, as well as those of the other two reviewers. In my opinion, the manuscript is now ready for publication.

Reviewer #2 (Remarks to the Author):

The authors of the manuscript entitled, “Quantifying soil accumulation of atmospheric mercury using fallout radionuclide chronometry” have responded to all of my comments and have made a clear effort to address all of the concerns raised. Although I think there are some questions still surrounding the mass balance aspect of their paper, I believe they now more clearly highlight areas of uncertainty in the use of their novel tool. As mentioned previously, I think this work is of high caliber and will be an important addition to the mercury literature. I thank the authors for their work and suggest acceptance of their manuscript for publication.

Below I have included a few minor edits the authors may choose to implement.

In many instances the citations occur out of order numerically.

Line 63: You may mention here that the extent to which re-emissions occur have an important effect on the surface lifetimes of mercury.

Line 130: I'd suggest removing “horizons” for readability.

Line 193: You might add the annualized value in brackets too, since up until now that has been the main format of discussion.

Line 339: add “the” to “...over past...”

Reviewer #3 (Remarks to the Author):

My concerns were fully met by the responses to the initial review

Response to Revision, NCOMMS-24-07443

We are pleased to satisfy all remaining suggestions of Reviewers and the Editor.

Reviewer #2 (Remarks to the Author):

The authors of the manuscript entitled, “Quantifying soil accumulation of atmospheric mercury using fallout radionuclide chronometry” have responded to all of my comments and have made a clear effort to address all of the concerns raised. Although I think there are some questions still surrounding the mass balance aspect of their paper, I believe they now more clearly highlight areas of uncertainty in the use of their novel tool. As mentioned previously, I think this work is of high caliber and will be an important addition to the mercury literature. I thank the authors for their work and suggest acceptance of their manuscript for publication.

Below I have included a few minor edits the authors may choose to implement.

In many instances the citations occur out of order numerically.

Final citations have been corrected.

Line 63: You may mention here that the extent to which re-emissions occur have an important effect on the surface lifetimes of mercury.

We have added this statement, " This uncertainty raises fundamental questions about the long-term efficacy of soils in sequestering legacy and contemporary Hg, **the lifetimes of Hg in the surface environment**, and as a result about the fate of Hg in a context of a changing global environment ^{1,3,4,6}.

Line 130: I'd suggest removing “horizons” for readability.

Revised accordingly.

Line 193: You might add the annualized value in brackets too, since up until now that has been the main format of discussion.

Values are annualized.

Line 339: add “the” to “...over past...”

Corrected.

Reviewer #3 (Remarks to the Author):

My concerns were fully met by the responses to the initial review